# Application of Metal Halide Perovskite in Internet of Things

**DOI:** 10.3390/mi15091152

**Published:** 2024-09-14

**Authors:** Zhihao Chai, Hui Lin, Hang Bai, Yixiang Huang, Zhen Guan, Fangze Liu, Jing Wei

**Affiliations:** 1Beijing Key Laboratory of Construction-Tailorable Advanced Functional Materials and Green Applications, Experimental Center of Advanced Materials, School of Materials Science and Engineering, Beijing Institute of Technology, Beijing 100081, China; 1120223734@bit.edu.cn (Z.C.); 3220241825@bit.edu.cn (H.L.); 1120212042@bit.edu.cn (H.B.); 1120230892@bit.edu.cn (Y.H.); 3120231039@bit.edu.cn (Z.G.); 2Advanced Research Institute of Multidisciplinary Sciences, Beijing Institute of Technology, Zhuhai 519088, China; fliu@bit.edu.cn

**Keywords:** Internet of Things, metal halide perovskite, sensor, indoor photovoltaic

## Abstract

The Internet of Things (IoT) technology connects the real and network worlds by integrating sensors and internet technology, which has greatly changed people’s lifestyles, showing its broad application prospects. However, traditional materials for the sensors and power components used in the IoT limit its development for high-precision detection, long-term endurance, and multi-scenario applications. Metal halide perovskite, with unique advantages such as excellent photoelectric properties, an adjustable bandgap, flexibility, and a mild process, exhibits enormous potential to meet the requirements for IoT development. This paper provides a comprehensive review of metal halide perovskite’s application in sensors and energy supply modules within IoT systems. Advances in perovskite-based sensors, such as for gas, humidity, photoelectric, and optical sensors, are discussed. The application of indoor photovoltaics based on perovskite in IoT systems is also discussed. Lastly, the application prospects and challenges of perovskite-based devices in the IoT are summarized.

## 1. Introduction

### 1.1. Basic Concepts of IoT

The Internet of Things (IoT) can collect various information about objects in real time by utilizing sensors and positioning systems. Physical and virtual objects can be seamlessly connected with various network accesses. The term ‘Internet of Things’ was initially proposed in the 1990s by Professor Kevin Ashton [1], a co-founder of the Auto-ID Laboratory at the Massachusetts Institute of Technology (MIT), to describe a system in which the internet is connected to the physical world via ubiquitous sensors. The IoT has garnered significant attention, and its applications span multiple fields including medicine, management, information technology, agriculture, and defense [2,3,4,5,6,7]. Recently, the IoT has tended to integrate with metaverse technology [8], an indispensable vehicle for the fourth industrial revolution [9].

### 1.2. Three-Layer Architecture of IoT

The IoT consists of three layers—a perception layer, a network layer, and an application layer [10]—as shown in Figure 1.

The perception layer covers various physical devices and sensors to gather physical data from the environment, serving as the source for IoT object identification and information collection.The network layer comprises various networks and computing platforms, functioning as the central hub of the entire IoT system. Data collected from the perception layer undergo further processing in this part.The application layer provides specific services to end users, acting as the interface between the IoT and users. This layer employs a variety of intelligent technologies to analyze and process data received from perception layers, which is closely integrated with industry requirements.

### 1.3. Development Status and Trend of Perception Layer

The IoT connects the physical and internet worlds via a series of sensors, which determine the upper limit of the IoT’s functionality [11]. Its further development depends on enhancing sensor functionality, endurance, and the application scope.

Regarding functionality, the variety and sensitivity of sensors are paramount. Sensors still need further development in sensitivity, working environment, and detection diversity to meet these demands.

Endurance capability is crucial for ensuring a continuous power supply for devices in the IoT [12], especially for those that are difficult to locate. Most of the IoT power supply relies on rechargeable batteries or wired power supply systems. Though rechargeable batteries can liberate sensors from scenario restrictions, the problems of battery endurance and replacement become particularly prominent when given an IoT system with millions of sensors.

The portability of IoT sensors is a concern as the IoT becomes more integrated into daily life with a diverse application scope. Many wireless, flexible, and wearable sensors have been developed, but they still face problems such as having lower sensitivity and stability than traditional devices and higher manufacturing costs [13,14,15].

The development of the IoT has raised higher demands for its sensors, energy sources, and device portability, which all rely on research on new materials.

In recent years, metal halide perovskite materials have attracted significant attention in the field of optoelectronics due to their tunable bandgap, excellent optoelectronic properties, low preparation cost, and strong process compatibility. These materials demonstrate enormous potential for development in the IoT domain. The tunable bandgap of perovskites enables broad-spectrum photoelectric conversion, making them suitable for manufacturing high-sensitivity light sensors. Perovskite sensors’ unique optical and chemical properties also allow them to accurately measure various environmental parameters. Additionally, perovskite is recognized as an outstanding photovoltaic material with high photoelectric conversion efficiency, which can serve as a solar cell to provide a power supply [16,17]. Furthermore, the flexibility, semi-transparency, and color adjustability of perovskite materials offer greater versatility in shaping sensor and battery designs. The solution-based fabrication method of perovskites in the production process results in higher compatibility and lower costs for large-scale applications [18,19,20,21,22].

There have been a considerable number of review papers that have separately covered perovskite-based gas and humidity sensors, photodetectors, and solar cells [23,24,25]. However, a comprehensive review focusing mainly on the applications of perovskite-based devices in the Internet of Things is lacking in the literature. In this review, we summarize the latest progress of perovskite in the IoT for its applications in sensors and photovoltaics. First, we introduce some basic concepts about perovskite, especially metal halide perovskite. Then, we summarize the recent progress in various perovskite-based sensors, including gas, humidity, photoelectric, and optical sensors. Furthermore, we introduce perovskite solar cells and indoor photovoltaic technology. Despite the considerable advancement in the field, there are several key challenges to its further development. Therefore, we provide a survey of the challenges and opportunities in the last section.

## 2. General Properties of Perovskites

### 2.1. Concepts

In 1839, Gustavus Rose, a mineralogist at the University of Berlin, discovered perovskite in the Ural Mountains of Russia and named the substance after the Russian geologist Lev Perovski [26]. Perovskite originally referred only to the material CaTiO_3_, but now more often refers to substances with perovskite’s structure and the formula ABX_3_ (A is an organic or inorganic metal ion, B is a metal ion, and X is an oxygen or halogen ion) [25]. 

### 2.2. Structure and Composition

This article primarily introduces metal halide perovskites (ABX_3_), where A generally refers to metals such as Cs or certain organic cations such as methylammonium (MA) and formamidinium (FA), while B typically denotes metals such as Pb. Under this premise, monovalent and divalent cations are usually stabilized at the A and B sites, respectively. A is situated at the center of the octahedron formed by X, as shown in Figure 2a, with a coordination number of 12. B is located at the center of the octahedron formed by X, as shown in Figure 2b, with a coordination number of 6. 

### 2.3. Properties

Metal halide perovskites have many special characteristics due to their unique structure. For example, when MAPbI_3_ is exposed to a specific gas environment (e.g., O_2_, NH_3_, or H_2_O), gas molecules can diffuse inside the perovskite crystal structure and reversibly fill iodine vacancies intrinsically present inside the crystals [27]. This phenomenon is called the ‘trap-healing mechanism’, which is beneficial for their application in gas and humidity sensors [28,29]. In addition, this material has a suitable bandgap of ∼1.55 eV, which matches the wavelength of ∼800 nm, making it a competitive light-absorbing material for solar cells and photosensors [24]. The bandgaps and spectra of perovskite materials can be adjusted by modulating the halogen component (I^−^, Br^−^, or Cl^−^) within the crystal structure, which greatly broadens the application scope of perovskite [30]. Moreover, perovskites possess other attractive properties, such as piezoelectricity, a high absorption coefficient (~105 cm^−1^), and a long charge diffusion length (>9.2 microns) [31,32], which make perovskites ideal candidates for sensors and optoelectronic devices in the IoT.

## 3. Perovskite Sensors

With the rapid expansion of the IoT, there is a growing demand for high-efficiency and low-energy devices. Against this backdrop, metal halide perovskite sensors, with characteristics including excellent gas sensitivity, humidity sensitivity, photoelectric properties, optical properties, self-powering capability, reversibility, flexibility, and high efficiency, are increasingly significant within the IoT landscape.

### 3.1. Gas and Humidity Sensor

Due to perovskite’s excellent electronic and ionic conductivity, perovskite gas sensors can detect very small changes in low-gas-concentration environments (the limit of detection is 8.85 ppm) [33,34] and enable efficient sensing at room temperature, which is an enormous advantage for gas sensing [35]. Conversely, traditional gas sensors, especially metal oxide semiconductor (MOS) sensors, still suffer from partial irreversibility [36] and require an operating temperature above room temperature to ensure that the surface activity can fully react with the target gas to obtain sufficiently high responses [37,38], which poses challenges for maintaining sensor stability and limits their application scope. 

The working principle of a perovskite gas sensor is based on the physical or chemical interaction between the sensing material and the target gas. These reactions change the electrical properties of the sensor, such as its resistance, capacitance, or conductivity, which depend on the properties of the target gases and semiconductor materials (n-type or p-type). In the case of n-type semiconductors, electron depletion regions appear on the material surface during chemisorption, causing an increase in resistance, while p-type semiconductors demonstrate the opposite behavior in terms of resistance changes because holes are the charge carriers [23]. Monitoring can be achieved by detecting such changes. 

The working principle of a perovskite humidity sensor is similar to that of a gas one. Water molecules can be easily adsorbed on the sensor’s surface due to their large dipole moment and affect the material’s properties. Devices made from perovskite were once infamous for failing due to H_2_O, which causes unsatisfactory long-term stability. However, good humidity sensors can be produced by taking advantage of perovskite’s excessive sensitivity to humidity [39].

In 2021, Li et al. [40] proposed an NH_3_-sensing mechanism in which the anomalous resistance enhancement is dominated by the grain boundaries of perovskites. They demonstrated that NH_3_ molecules can substitute the MA^+^ cations of MAPbI_3_ to form insulating NH_4_PbI_3_·MA intermediate layers on the surface of the crystal grains, increasing the resistance, as shown in Figure 3a. Additionally, they constructed an MAPbI_3_-based gas sensor and achieved a gas response of 472% toward 30 ppm of NH_3_. This study guides the development of high-performance sensing perovskite materials.

In the same year, Wu et al. [41] demonstrated an excellent impedance relative humidity sensor based on all-inorganic halide perovskite CsPbBr_3_ nanoparticles (NPs) under a low working voltage (20 mV). The CsPbBr_3_ NP humidity sensor exhibits fast response and recovery behavior at room temperature. As shown in Figure 3b,c, when the sensor switches in different humidity environments, the impedance of the metal halide perovskite sensor (CsPbBr_3_) shows a faster spring back (60 s) than traditional ones (ZnO/SnO_2_ 90s) [42]. Meanwhile, the impedance of the perovskite sensor is more stable than that of traditional ones when the RHs and operating frequencies change, as shown in Figure 3d,e. In addition, the CsPbBr_3_ NP humidity sensor produces low hysteresis outputs (3%) when measuring different RH levels with upward and downward trends, further demonstrating good reversibility. The study suggests that humidity sensors made of metal halide perovskite, such as CsPbBr_3_ NPs, have great potential in future real-time monitoring applications, especially the IoT.

In 2022, Cho et al. [43,45] first chose all-inorganic metal halide perovskites to prepare humidity sensors. They successfully compounded CsPbBr_3_ and CsPb_2_Br_5_ with various traditional humidity-sensitive ceramic materials (Al_2_O_3_, TiO_2_, and BaTiO_3_) using the aerosol deposition method and made a novel capacitive humidity sensor. As shown in Figure 3f,g (where RH is the relative humidity), the CsPbBr_3_-impregnated humidity sensor shows remarkably high sensitivity. In 2024, Zhang et al. [46] introduced a methylamine gas sensor based on MAPbBr_3_ and the aggregation-induced emission (AIE) material (S)-OBN-tCz. Incorporating AIE materials has successfully improved the signal intensity of optical sensors through the Förster resonance energy transfer mechanism and passivation effect, thereby effectively enhancing the sensitivity of sensors. These studies show that the effectiveness and sensitivity of sensors can be boosted by combining other materials with lead halide perovskite, providing an efficient way to further improve existing sensors.

Although metal halide perovskite sensors have many advantages, the most used lead halide perovskite faces the problem of lead toxicity. Long-term exposure to lead causes irreversible physical damage to the human body, which limits metal halide perovskite sensors’ application in the IoT, especially family applications [47]. In 2024, Maity et al. [44] reported a lead-free halide perovskite-based gas sensor to detect ammonia gas. The sensor uses methylammonium tin iodide (CH_3_NH_3_SnI_3_ or MASnI_3_ or MASI) as the active material. The maximum calibrated sensitivity based on the electrical readout of the sensor is ∼85% upon exposure to 100 ppm of ammonia gas, as shown in Figure 3h. The sensor can be operated at a 2 V bias with an output current of ∼2 nA, making the device compatible with low-power e-textile-based wearable gas sensors. Furthermore, the gas-sensing performance of the lead-free halide perovskite-based ammonia sensor has been compared with its lead-based counterpart, as shown in Figure 3i (where MAPI stands for MAPbI_3_, and MASI stands for MASnI_3_). Though the lead-free halide perovskite-based gas sensor has a lower sensitivity, the study shows that the Pb element can be changed into other elements to avoid lead toxicity so the sensor can be applied to family life, which lays a foundation for the further application of metal halide perovskite sensors in the IoT. 

### 3.2. Photodetector

Photodetectors (PDs) operate based on the transition of electrons from a lower-energy state to a higher-energy state under photonic illumination. PDs can be divided into photoconductive (PC)-mode and photovoltaic (PV)-mode detectors. PC-mode detectors identify a light signal by detecting the change in resistance across the photoactive material under illumination. In contrast, PV-type PDs utilize the built-in electric field in heterojunctions or Schottky junctions to separate the photogenerated electron−hole pairs. Thus, there is no need for an external bias, so the PD can become a self-powered device [24]. Figure 4a,b illustrate the charge separation under the PV effects in a heterojunction and a Schottky junction.

The most common photodetectors are made from inorganic semiconductor materials. However, the slow response times observed in these devices severely restrict their application in high-speed devices [48]. Given these issues, the application of metal halide perovskite materials in photodetectors has attracted attention. Metal halide perovskite’s main inherent defects are shallow defects that do not affect the rapid response of optoelectronic sensors. Therefore, utilizing metal halide perovskite in photodetectors can effectively suppress dark current, increase the linear dynamic range, and achieve high detection rates and fast responses [49,50,51], which makes metal halide perovskite PDs widely used in wavelength selecting, optical imaging, and optical communications [52,53]. Furthermore, perovskite’s excellent mechanical properties also make it possible to prepare PDs on a flexible platform [54], which greatly expands its application in the IoT.

**Figure 4 micromachines-15-01152-f004:**
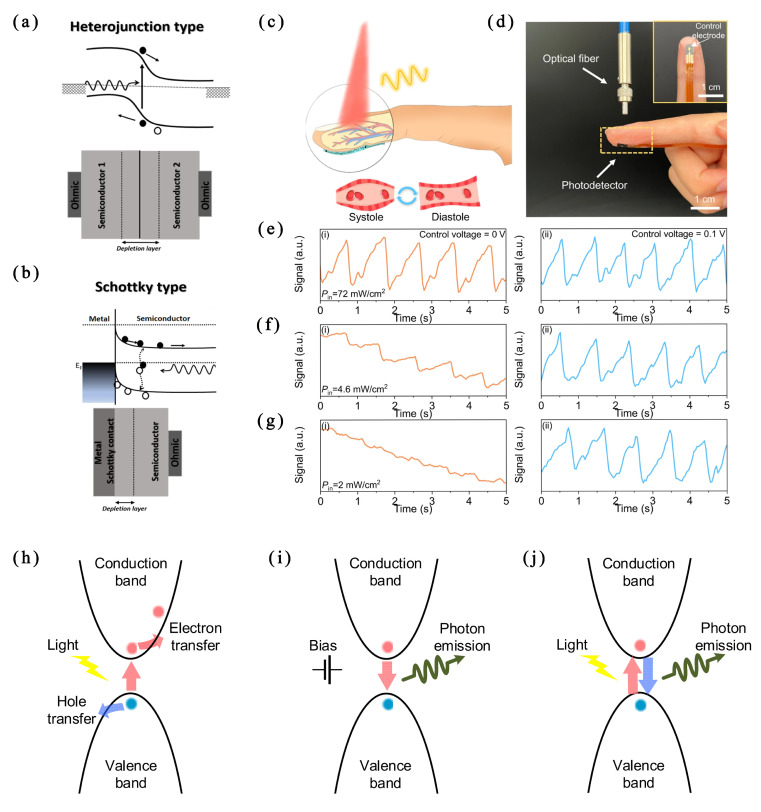
Schematic diagrams of the working principle of self-powered PDs in PV mode: (**a**) heterojunction type; (**b**) Schottky type [24]. Application of the FPDs for blood pulse signal detection: (**c**) schematic diagram of the working principle of the photoplethysmography test. Volumetric changes in the blood vessels modulate the transmitted light intensity. (**d**) Photograph of the FPD attached on finger pulp as a photoplethysmography sensor for recording blood pulse signals. An 800 nm light beam is generated from the optical fiber and reaches the FPD through the finger. The inset shows the top-view photograph of the FPD connected with a flexible printed circuit board. (**e**–**g**) Comparison of photoplethysmography signals detected by the FPD under different incident light intensities (72, 4.6, and 2 mW/cm^2^) when the control electrode was applied with (i) 0 V and (ii) 0.1 V. The calculated blood pulse frequency was 67 beats per minute [55]. (**h**) Photoelectric, (**i**) electro-optical, and (**j**) all-optical conversion mechanisms of perovskite [56].

In 2023, Algadi et al. [57] produced a self-powered and high-performance vertical-type heterojunction photodetector based on solution-processed NGQDs/CsPbBr_3_. The PD exhibited a high performance with a light current of 257.71 nA, an on/off ratio of 670, a responsivity R = 3.21 A/W, a specific detectivity D* = 2.9 × 1012 J, and an external quantum efficiency EQE = 270% under the illumination of a light source with a wavelength of 520 nm and a power intensity of 0.8 mW/cm^2^ at a bias voltage of 3 V. It was confirmed to operate without an external bias voltage, demonstrating the device’s photovoltaic characteristics at 0 V. Miao et al. [58] prepared a compact, uniform, pinhole-free, and highly crystallized 2D−3D gradient perovskite film (MAPbI_3_) using the hot-casting method. They produced a photodetector with an extremely low dark current density (2.3 × 10^−11^ A/cm^2^), a high specific detectivity (1.22 × 1014 Jones) at 455 nm, an ultra-fast response time (5.5/4.7 µs), and an ultra-high on/off ratio of 108. These studies indicate metal halide perovskite’s great potential for high-efficiency photodetectors without a voltage bias. 

In 2024, Tang et al. [55] proposed an electrical field modulation strategy to significantly reduce the dark current of a metal halide perovskite-based (MAPbI_3_) flexible photodetector by more than 1000 times (from ~5 nA to ~5 pA). Meanwhile, ion migration in metal halide perovskites is effectively suppressed, and the photodetector shows a long-term continuous operational stability (~8000 s) with low signal drift (~4.2 × 10^−4^ pA per second) and ultra-low dark current drift (~1.3 × 10^−5^ pA per second). Benefitting from the excellent light-sensing performance, flexible photodetectors (FPDs) prepared via this strategy can obtain high-fidelity blood pulse waveforms under a low incident light intensity (2 mW/cm^2^), as shown in Figure 4c–g, which is critical for their application in low-power-consumption wearable electronics. This work offers a universal strategy to improve the performance of metal halide perovskites for wearable flexible photodetectors, showing the application prospect of metal halide perovskites in flexible IoT devices.

### 3.3. Optical Conversion Sensor

Perovskite materials have excellent photovoltaic, electro-optical, and all-optical conversion properties compared with traditional sensors. Therefore, various types of perovskite optical conversion sensors can be developed with these properties.

The mechanism of photo/bias regulating charge carrier (electron−hole) generation and transfer is shown in Figure 4h–j. With good optoelectronic properties, metal halide perovskite can capture incident light and generate excitons (electron−hole pairs). These excitons can overcome the exciton binding energy and separate into free electrons and holes, i.e., undergo ‘photoelectric conversion’, as shown in Figure 4h. Excited by the external bias voltage, the electrons in the higher-energy state of the conduction band recombine with the holes in the valence band and emit fluorescence photons, in a process called ‘electro-optical conversion’, as shown in Figure 4i. Meanwhile, all-optical conversion occurs when metal halide perovskite is illuminated, as shown in Figure 4j. Under an overloaded external bias and illumination, the ions in the perovskite (I^−^, Br^−^, Pb^2+^, and MA^+^) and electrode (Ag^+^) migrate out of the ABX_3_ perovskite structure, resulting in defect band levels, and the resistance changes subsequently [56]. 

In 2020, Chen et al. [59] engineered a super-hydrophilic CsPbBr_3_@polystyrene/polyacrylamide all-optical conversion sensor that can quantify Fe^3+^ levels in human blood. Beyond Fe^3+^, these sensors exhibit sensitivity toward various metabolic substances in the human body, such as urea, chloride ions, and iodide ions. Wang et al. [60] utilized the photo-luminescence property of CsPbBr_3_ quantum dots and the hydrophobicity of boron nitride to construct a hydrophobic fluorescence sensor for tetracycline identification. The linear detection range of the sensor was 0–0.44 mg·L^−1^, and the detection limit was as low as 6.5 ng·mL^−1^. Xiang et al. [61] utilized the easy degradability of CsPbBr_3_ in water to develop a highly humidity-sensitive photo-luminescence sensor, which can detect the water content in traditional Chinese medicine. The sensor showed a good linear relationship within the relative humidity range of 33–98%. The relative humidity detection limit was 12%. 

These studies show that various perovskite sensors can be developed by using the material’s optical properties.

### 3.4. Nanogenerator

Besides self-powered photodetectors, sensors lacking this self-powering ability can be made via another method. A nanogenerator (NG) is a new technique, first proposed by Wang et al. [62] in 2007, that utilizes mechanical and thermal energies produced by human body motion and then converts them into electrical energy. In 2019, Wang et al. [63] first predicted and proposed the friction volt effect. In 2020, Zhang et al. [64] experimentally verified this phenomenon and defined the ‘friction volt effect’ as the generation of direct current at the interface between a metal/semiconductor and another semiconductor due to friction. After that, researchers discovered that using the ‘friction volt effect’ to convert mechanical energy into electrical energy enables self-powered sensors [65]. Perovskite materials have excellent semiconductor properties. They can generate a direct current signal and separate the photoelectric electrons and holes by forming a heterojunction [66]. Therefore, perovskite combines the photoabsorption characteristics and piezoelectric/friction effect and can be a suitable candidate for self-powered sensors [67].

In 2019, Sultana et al. [68] developed a piezoelectric−pyroelectric nanogenerator (PPNG) based on a MAPBI_3_−polyvinylidene fluoride material that can harvest mechanical and thermal energies. Applying a periodic compressive contact force at a frequency of 4 Hz generates an output voltage of ∼220 mV. The PPNG has a piezoelectric coefficient (d_33_) of ∼19.7 pC/N coupled with a high durability (60,000 cycles) and quick response time (∼1 ms). The maximum generated output power density (∼0.8 mW/m^2^) is sufficient to charge up various capacitors.

Subsequent research on NGs has focused on improving the band alignment, carrier recombination, and mechanical wear between interfaces [69]. Though integrating NGs with sensor devices is challenging, the further development of NG-based self-powered sensors is attractive.

## 4. Perovskite Solar Cells and Indoor Photovoltaics

### 4.1. Research Progress

The primary power source for wireless sensor devices in the IoT is mainly rechargeable batteries. However, these batteries require regular maintenance and replacement and pose certain environmental hazards, which are not conducive to green and sustainable practices. Therefore, the technology of harvesting and collecting energy from natural sources such as light, heat, and magnetism has garnered attention from researchers. Among these technologies, solar cell technology stands out as it enables low-energy wireless portable devices to sustain their own power supply without the need for replacement or maintenance.

The development of solar cells can be categorized into three generations: silicon-based solar cells (first generation), thin-film solar cells (second generation), and various new types of solar cells with higher efficiency (third generation). Compared with the first two generations, third-generation solar cells offer lower costs and higher efficiency while experiencing rapid development. As depicted in Figure 5a, the power conversion efficiency (PCE) of perovskite solar cells (PSCs) increased from 14.1% in 2013 to 26.7% by 2024 [70]. Consequently, PSCs hold promising prospects for future development [71].

### 4.2. Advantages

#### 4.2.1. Excellent Photoelectric Property

Compared with traditional solar cells, which can only absorb a fixed spectrum of light, perovskite can ensure that the maximum light sensitivity of the cell pairs well with its light source spectrum, making it an ideal choice for self-powered and sustained operation in IoT sensors. Among various perovskite types, halide perovskites stand out due to their superior adjustable bandgap, bipolar charge transport characteristics [74], strong light absorption, long carrier lifetime, and solution processability. They exhibit outstanding performance in outdoor environments and achieve a conversion efficiency of over 40% in indoor photovoltaics (IPVs) [73,75].

As shown in Figure 5a, the tunable bandgap of perovskite leads to exceptional photovoltaic conversion performance. In current laboratory research on PSCs, the highest photovoltaic conversion efficiency has reached 26.1%, approaching the maximum efficiency of silicon cells of 26.7% [70].

Furthermore, perovskite demonstrates a lower carrier density and recombination probability along with a higher charge extraction efficiency in low-light-intensity indoor environments. Therefore, the built-in electric field can drive carriers to move faster and generate a high fill factor (FF), as shown in Figure 5c. The open-circuit voltage and photovoltaic conversion efficiency of indoor photovoltaic PSCs are higher than those of other materials [72].

#### 4.2.2. Low Cost and Simple Preparation Process

Compared with other mature and emerging photovoltaic technologies, the manufacturing process of PSCs is simple and economical. PSCs have lower energy consumption with high solubility at temperatures below 100 °C [76].

Silicon-based solar cells often require high-temperature environments and special manufacturing processes (such as texturing and anti-reflection coatings [77]). Meanwhile, a perovskite thin film can be deposited using various simple processes, such as spin coating, printing, vapor-assisted solution deposition, thermal vapor deposition, inkjet printing, slit-die coating, and spray coating [78], and it is possible to improve perovskite’s water stability by using some specific preparation process, including the substitution of A cations, ligand exchange, encapsulation in porous frameworks, passivation with inorganic or organic layers, and encapsulation in hydrophobic polymers and glass matrices [79,80,81,82].

Taking the new form of lead-based halide perovskite nanocrystals (NCs) as an example, the synthetic methods for NCs are relatively simple, mainly using a method called ligand-assisted reprecipitation (LARP), which offers NCs size tunability via varying the precursor composition and reaction temperature [83]. LARP combines the processes of reprecipitation and supersaturation recrystallization, mixing the perovskite precursor solution with hexane and toluene at low temperatures to create a highly supersaturated non-equilibrium state in the entire liquid system. This promotes the rapid precipitation of metal halide salts, i.e., the nucleation and growth of perovskite NCs, which is a simple and effective method [84].

#### 4.2.3. Excellent Mechanical Properties

PSCs exhibit outstanding mechanical properties, including light weight, flexibility, and transparency. As a result, PSCs have significant potential for applications in portable, bendable, and wearable devices compared with silicon-based solar cells, particularly in ultra-thin devices, such as wearable sensors in the IoT and flexible electronic products [85].

### 4.3. Applications of PSCs in IoT Field

The application of solar cells in the IoT field primarily serves as a power supply module for various sensors and devices. Currently, the development of conventional full-spectrum photovoltaic technology is relatively mature and has demonstrated excellent performance under conditions of ample outdoor illumination. However, progress in indoor photovoltaic technology has been relatively slow compared with outdoor applications. This is mainly because indoor environmental light typically comes from sunlight through windows or artificial lighting equipment. The spectral characteristics of these light sources significantly differ from outdoor sunlight, resulting in a low energy density and limited wavelength range. Therefore, effective methods for utilizing photovoltaic power as an indoor power supply are lacking.

Indoor photovoltaic (IPV) technology is generally defined as the application of photovoltaics under low-incident-light-intensity conditions. Commonly used indoor light sources, such as white light-emitting diodes (LEDs) and fluorescent lamps, come with a minimal amount of ambient diffuse light within a spectral range of 400–750 nm and illuminance levels ranging from 200 to 1000 lux [86,87]. The power density is lower than that provided by natural light sources. Hence, the design considerations for indoor photovoltaics significantly differ from those applied to outdoor systems.

However, the issue of PSCs’ low resistance to humidity or other gases is insignificant due to the mild indoor environmental conditions with no UV radiation, lower humidity, reduced thermal stress, and absence of other harmful gases. Furthermore, their outstanding performance in indoor photovoltaics has positioned them as an ideal power supply method for indoor IoT sensors [88]. The adjustable bandgap of lead halide perovskite solar cells can perfectly match the spectral distribution of indoor light sources and effectively power indoor IoT devices [89]. Lead halide perovskite solar cells can simultaneously collect direct, diffused, and reflected light to achieve a higher photoelectric conversion efficiency via optimization [90], meeting the power requirements of various low-power electronic devices and sensors [91]. Additionally, their manufacturing technique is relatively simple compared with that of silicon-based solar cells, which is crucial for the large-scale deployment of sustainable energy in IoT systems [92].

In 2024, Ma et al. [93] reported on wide-bandgap PSCs to realize 44.72%-efficient indoor photovoltaics. The incorporation of a trace amount of dual additives enabled a high-quality and less-defective wide-bandgap perovskite film with mitigated halide segregation, leading to the suppression of bulk trap-induced nonradiative recombination losses. The NiO_x_-based inverted champion cell under one-sun illumination generated a record power conversion efficiency (PCE) of 21.97%, an impressive FF of 83.4%, a J_sc_ of 21.56 mA/cm^2^, and a V_oc_ of 1.25 V for 1.71-electronvolt wide-bandgap perovskites, as shown in Figure 6. Such devices also showed high operational stability over 800 h during T95 lifetime measurements. The study proved that wide-bandgap perovskite, such as metal halide perovskite, is a promising material for making indoor PSCs.

Furthermore, PSCs can be made foldable and stretchable due to their excellent mechanical properties, making them applicable to various fields such as e-paper and electronic clothing. Simultaneously, various anti-impact, highly flexible special devices can also be achieved, such as flexible perovskite-based electronic shelf labels (PESLs), portable electronic products, textile electronic integration devices, and transportation units [94,95].

As shown in Figure 7, IoT-enabled devices will also play a crucial role in full lifecycle monitoring scenarios, such as smart homes and green factories [96]. Perovskite indoor photovoltaics can effectively solve the problem of a sustained power supply for large IoT systems, providing these systems with green and sustainable clean energy.

## 5. Conclusions and Prospects

The Internet of Things (IoT), as a further development of the internet, has become an indispensable future development trend. However, the perception layer, a key component of the IoT, is limited by traditional materials and needs to be further developed. Furthermore, with the rapid advancement of 5G communication technology, there is an increasing demand from both enterprises and individuals for a greater number of sensors, which poses a challenge for sensors’ power supply and portability.

Metal halide perovskite, a novel type of semiconductor material, has garnered extensive attention in the fields of gas sensors, humidity sensors, photodetectors, optical conversion sensors, nanogenerators, and indoor photovoltaics due to its outstanding photoelectric conversion properties, adjustable and easily processable energy level structure, and controllable crystal structure and morphology. This highlights its potential application in the IoT domain. Moreover, by combining metal halide perovskite with other materials, more efficient and non-toxic devices can be developed. Metal halide perovskite’s application in nanogenerators illustrates its potential as a self-powered device material, which is important for large-scale sensors in the IoT. Finally, the huge advantages of metal halide perovskite compared with traditional photovoltaic materials in indoor photovoltaics enable it to excel in the IoT realm as solar cells. However, metal halide perovskite sensors still have many shortcomings, such as poor stability and toxicity, which limit their further application.

Based on the present research, metal halide perovskite materials will frequently feature in future IoT applications, and better performance will be achieved in future developments.

Despite the promising prospects for perovskite sensors in IoT applications, it is essential to continue addressing some technical and practical challenges that arise during their implementation, as follows.

First, the long-term stability of PSCs is relatively weak. Studies have shown that perovskite materials are unstable when exposed to environmental factors such as oxygen, water, heat, and light, as they induce phase transitions and lattice strains in halide perovskites [97,98]. In addition, some studies have shown that the intrinsic properties of perovskite materials, such as ion migration and low defect formation energy, play an important role in promoting the rapid decomposition of perovskite thin films, leading to PSCs that cannot meet the market requirement of a 25-year lifespan [99]. In this case, the following solutions can be undertaken: adjusting the proportion of the perovskite components to increase the proportion of bromine and iodine; completely or partially replacing the high-volatility A-site MA cation with FA or Cs; using 2D material polymers and fullerene derivatives to passivate the defects in perovskite thin films; increasing the grain size to avoid the appearance of PbI_2_; and forming an ordered dipolar structure to reduce the influence of heat [99,100].

Second, the leakage of toxic lead components causes potential health risks. The lead leakage from mixed halide perovskite into the environment may cause potential dangers. Therefore, lead capture in the encapsulation, charge transport, and perovskite layers, as well as the overall device-level packaging and end-of-life recycling, need to be considered [101,102,103].

Third, it is difficult to control the quality of perovskite thin films. Although their conversion efficiency under indoor lighting has been greatly improved in recent studies, a large gap with the Shockley−Queisser theoretical efficiency remains. This is mainly because perovskite thin films prepared via the widely used reverse solvent method cannot obtain high-quality perovskite thin films with micron thickness, resulting in serious light loss, low short-circuit current density, and serious nonradiative recombination loss, further reducing the open-circuit voltage and fill factor. Therefore, it is urgently necessary to study new thin-film preparation methods or reduce the defects generated during thin-film preparation to increase the film thickness, regulate the film bandgap, and improve the film quality, thus improving indoor photovoltaic devices’ performance.

Fourth, there is no uniform indoor spectral standard. A uniform indoor spectral standard needs to be established to promote subsequent research and production so that indoor photovoltaics’ performance indicators can be tested.

## Figures and Tables

**Figure 1 micromachines-15-01152-f001:**
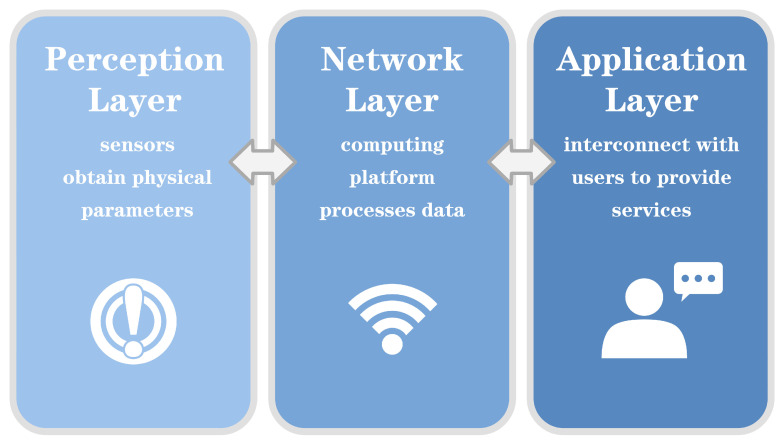
Diagram of the basic structure of the Internet of Things.

**Figure 2 micromachines-15-01152-f002:**
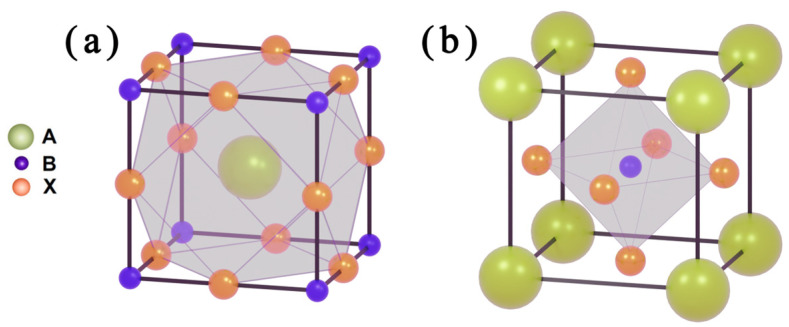
Schematic diagram of perovskite’s structure: (**a**) coordination number is 12; (**b**) coordination number is 6.

**Figure 3 micromachines-15-01152-f003:**
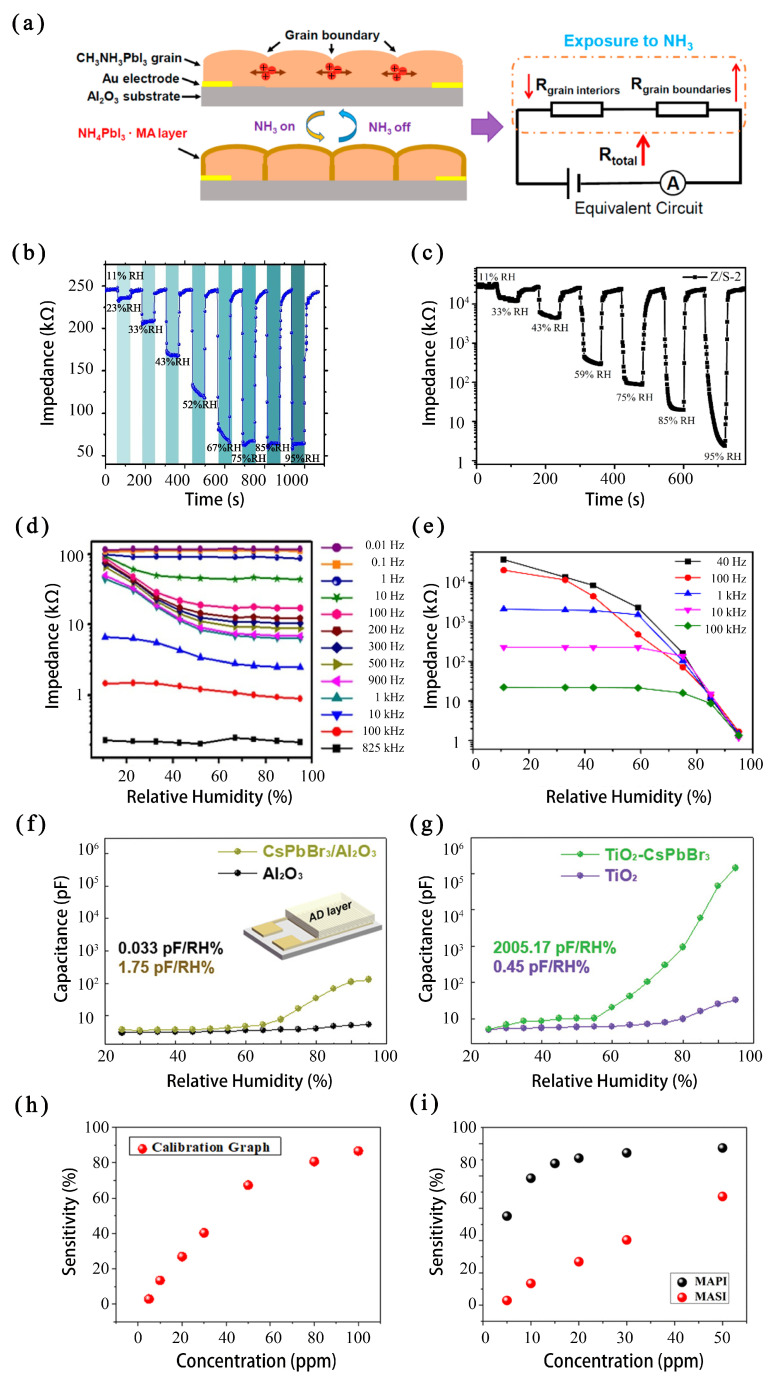
(**a**) Illustration of an NH_3_-sensing mechanism in MAPbI_3_ film [40]. Impedance response of the humidity sensor at different RHs: (**b**) CsPbBr_3_; (**c**) ZnO/SnO_2_. Impedance response of the humidity sensor at different operating frequencies: (**d**) CsPbBr_3_; (**e**) ZnO/SnO_2_ [41,42]. Capacitance variation with different RHs ranging from 25 to 95 RH%: (**f**) Al_2_O_3_ and CsPbBr_3_/Al_2_O_3_; (**g**) TiO_2_ and CsPbBr_3_/TiO_2_ [43]. (**h**) Sensitivity of the lead-free halide perovskite-based gas sensor at different NH_3_ concentrations. (**i**) Comparison of sensitivity between lead halide MAPbI_3_ (MAPI) and lead-free halide MASnI_3_ (MASI) perovskite NH_3_ sensors both grown on a fabric substrate [44].

**Figure 5 micromachines-15-01152-f005:**
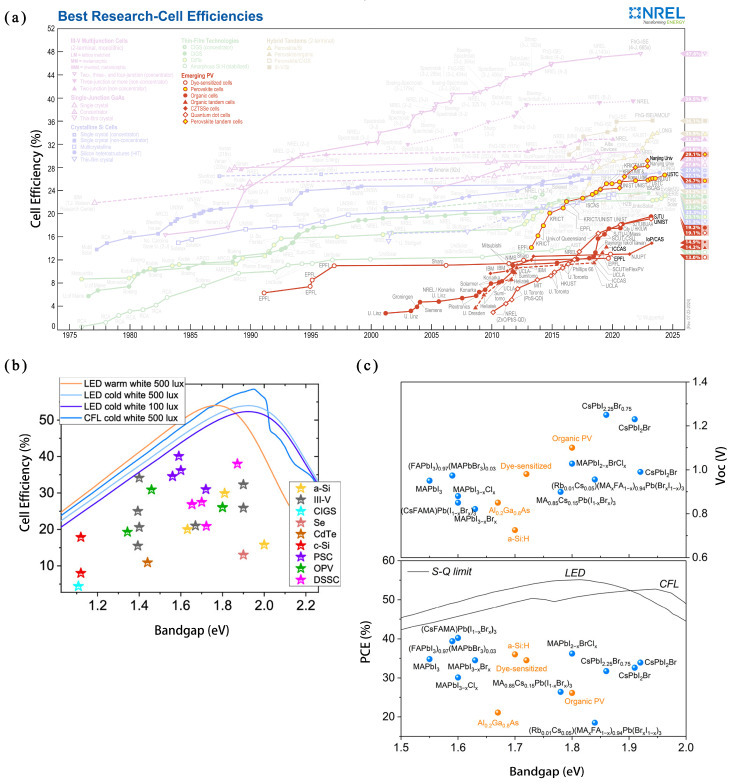
(**a**) Development of perovskite solar cells’ PCE [70]. (**b**) Functional relationship between performance and bandgap of several photovoltaic technologies when indoor light is between 50 and 3000 lx [72]. (**c**) Comparison of PCE and V_OC_ between perovskite IPV and other IPVs [73].

**Figure 6 micromachines-15-01152-f006:**
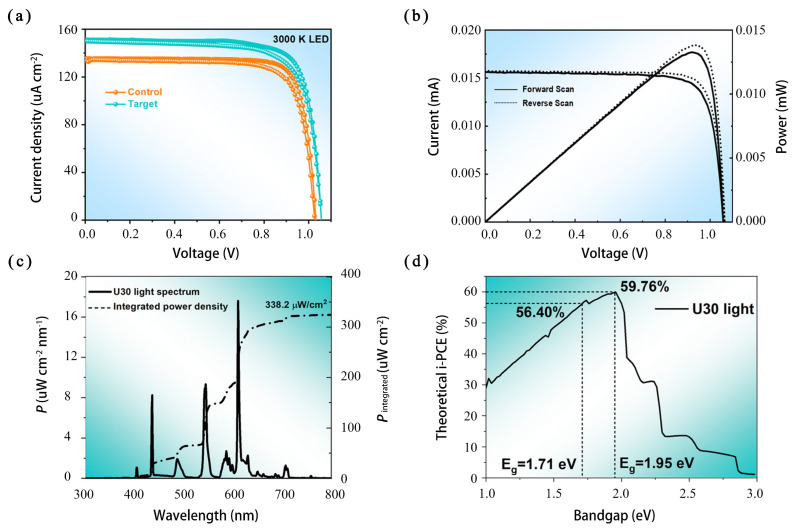
Indoor photovoltaic performance of wide-bandgap PSCs: (**a**) current density-voltage (J-V) curves of control and target wide-bandgap PSCs under LED light with 1000 lux and 300 K; (**b**) indoor current-voltage (I-V) and corresponding spectrum distribution with an integrated power density of 338.2 mW/cm^2^ under U30 light of 1000 lux; (**c**,**d**) U30 light spectrum distribution with an integrated power density of 338.2 mW/cm^2^ and the corresponding Shockley-Queisser limit of PCE of an ideal photovoltaic device as a function of bandgap energy [93].

**Figure 7 micromachines-15-01152-f007:**
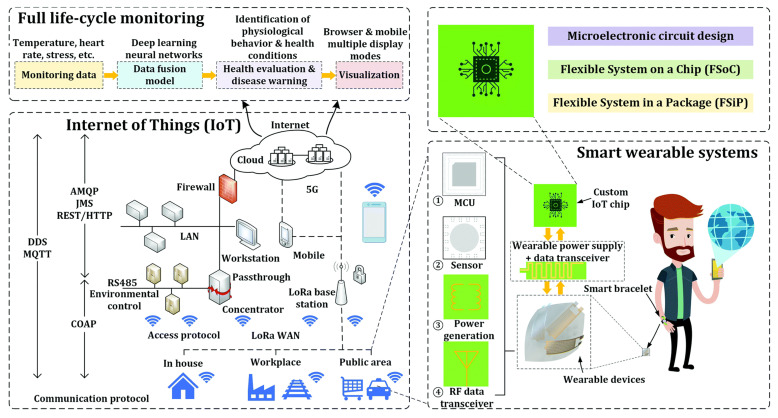
Standalone IoT-enabled systems for full lifecycle health monitoring. Depending on the application scenario, the system can be powered by a green and sustainable power source [96].

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
