# Peer review of "Application of Metal Halide Perovskite in Internet of Things"

_micromachines, 2024, doi:10.3390/mi15091152_

Round 1

Reviewer 1 Report

Comments and Suggestions for Authors

In the manuscript titled "Application of metal halide perovskite in Internet of Things," the authors review the use of metal halide perovskites in IoT systems, particularly in sensors and energy supply modules. They highlight recent advancements in perovskite-based sensors, including gas, humidity, photoelectric, and optical sensors, and explore the potential of indoor photovoltaics for IoT. The unique properties of metal halide perovskites, such as excellent photoelectric performance, adjustable bandgap, flexibility, and mild processing, demonstrate their potential to address current IoT device limitations, like high-precision detection and long-term endurance.

The manuscript is well-structured and provides valuable insights into the application prospects and challenges of perovskite-based devices in IoT. The methodologies discussed for incorporating perovskites into IoT components are innovative and show promise for enhancing the performance of IoT systems. Therefore, I recommend this manuscript for publication in the relevant journal after minor revisions. The minor comments for this manuscript are as follows:

1.      While the manuscript mentions various types of sensors (gas, humidity, photodetector, and optical), it would benefit from a more detailed explanation of the mechanisms by which metal halide perovskites enhance sensor performance.

2.      The manuscript briefly mentions the limitations of traditional materials but does not provide a direct comparison with perovskites. Including a table or figure comparing key performance metrics between traditional materials and metal halide perovskites would improve the reader's understanding.

3.      The manuscript lacks a conclusion. The authors are advised to include a conclusion to summarize their findings and suggest future research directions.

Reviewer 2 Report

Comments and Suggestions for Authors

In the review article, Chai et al. represented the use of perovskites in IoT applications beautifully. Here are my comments on the articles below-

1. On page no 2, line no 50: It will be IoT instead of LoT I guess.

2. On page no 3 line no 95-97: “As early as 1839, Gustavus Rose, a mineralogist at the University of Berlin, discovered perovskite in the Ural Mountains of Russia and named the substance after Russian geologist Lev Perovski.” Please add proper reference.

3. On page no 4 line no 124: please put the value of the absorption coefficient and carrier diffusion length.

4.  On page no 4 line no 135: Please provide the value of “ low gas concentration environments”.

5. On page no 4 line no 155-157: Please put proper references on “Devices made from perovskite were once infamous for failing due to H2O, which causes unsatisfactory long-term stability. But by taking advantage of perovskite’s excessive sensitivity to humidity, it is possible to produce quite good humidity sensors.”

6. Why lead free halide perovskite have higher sensitivity over lead-based perovskite? Please explain it.

7. Please explain more about the stability of halide perovskites and how to encounter the toxicity issue for real-life use in detail in the revised manuscript.

8. On page no 9 Figure 5(b) is not updated please update it in the revised manuscript.

9. The authors did not explain about the synthesis of perovskites in their review article. Authors are suggested to add this section in the revised manuscript. Authors can explain these articles in their revised manuscript:- “10.1039/D2TA09286G, doi.org/10.1021/acsnano.3c05609, doi.org/10.1016/j.mtphys.2023.101079, doi.org/10.1002/ange.202302852, doi.org/10.1021/acsomega.9b00829”

10. On page no 13 line no 453-458: Authors are suggested to put proper references in this section.

11. There are lots of typos like in detectivity, signal drift, responsivity and so on. Authors are suggested to revise the article thoroughly. Also, the language can be improved in the revised manuscript.

12. Also authors are strongly suggested to update their references in the revised manuscript.

13. Authors can add or explain some of the recent reports in the revised manuscripts:- “doi.org/10.1002/adfm.202202087, 10.1126/science.add7331, doi.org/10.1002/aenm.202001305, doi.org/10.1002/smtd.202201499, doi.org/10.1002/adfm.202311205, doi.org/10.1002/adom.202300233, doi.org/10.1021/acsenergylett.3c01649”.

Comments on the Quality of English Language

Language can be extensively modified.

Round 2

Reviewer 2 Report

Comments and Suggestions for Authors

In the paper “Application of metal halide perovskite in the Internet of Things” Chai et al. carefully revised the manuscript according to suggestions. Here are some more comments-

1. In Figure 3, the x-y-axis legends are not in the same format. Please make it the same.

2. There are still some superscript and subscript issues. Please read carefully and make it appropriate.

3. Figure 4 h, I and j resolution can be improved.

4. In Figure 6, the x-y-axis legends are not in the same format. Please make it the same.

5. The resolution of Figure 7 must be improved a lot. It is very difficult to read the authors.
